# βPix Guanine Nucleotide Exchange Factor Regulates Regeneration of Injured Peripheral Axons

**DOI:** 10.3390/ijms241814357

**Published:** 2023-09-20

**Authors:** Yewon Jeon, Yoon Kyung Shin, Hwigyeong Kim, Yun Young Choi, Minjae Kang, Younghee Kwon, Yongcheol Cho, Sung Wook Chi, Jung Eun Shin

**Affiliations:** 1Department of Life Sciences, Korea University, Seoul 02841, Republic of Korea; wjsdp@korea.ac.kr; 2Peripheral Neuropathy Research Center (PNRC), Department of Molecular Neuroscience, College of Medicine, Dong-A University, Busan 49201, Republic of Korea; sykpooh@dau.ac.kr (Y.K.S.); gnldrud123@naver.com (H.K.); mirine0224@dau.ac.kr (Y.Y.C.); ine_b_@donga.ac.kr (M.K.); 3Department of Translational Biomedical Sciences, Graduate School of Dong-A University, Busan 49201, Republic of Korea; 4Department School of Biological Sciences, Seoul National University, Seoul 08826, Republic of Korea; 5Department of Brain Sciences, DGIST, Daegu 42899, Republic of Korea; axon@dgist.ac.kr

**Keywords:** guanine nucleotide exchange factor, Rac1, small GTPase, growth cone, axon regeneration, dorsal root ganglion, sciatic nerve injury, Src

## Abstract

Axon regeneration is essential for successful recovery after peripheral nerve injury. Although growth cone reformation and axonal extension are crucial steps in axonal regeneration, the regulatory mechanisms underlying these dynamic processes are poorly understood. Here, we identify βPix (Arhgef7), the guanine nucleotide exchange factor for Rac1 GTPase, as a regulator of axonal regeneration. After sciatic nerve injury in mice, the expression levels of βPix increase significantly in nerve segments containing regenerating axons. In regrowing axons, βPix is localized in the peripheral domain of the growth cone. Using βPix neuronal isoform knockout (NIKO) mice in which the neuronal isoforms of βPix are specifically removed, we demonstrate that βPix promotes neurite outgrowth in cultured dorsal root ganglion neurons and in vivo axon regeneration after sciatic nerve crush injury. Activation of cJun and STAT3 in the cell bodies is not affected in βPix NIKO mice, supporting the local action of βPix in regenerating axons. Finally, inhibiting Src, a kinase previously identified as an activator of the βPix neuronal isoform, causes axon outgrowth defects in vitro, like those found in the βPix NIKO neurons. Altogether, these data indicate that βPix plays an important role in axonal regrowth during peripheral nerve regeneration.

## 1. Introduction

Peripheral nerve axons are more frequently exposed to mechanical insults than their counterparts in the central nervous system, owing to the lack of skeletal protection. However, even after severe axonal damage, such as complete cutting of axons, peripheral neurons can repair themselves with a remarkable ability to regrow damaged axons from the lesion site [1,2,3]. Regenerating peripheral axons can reach the target tissue, such as cutaneous receptors and muscle fibers, rebuild synapses, and finally reconstruct neural connections [4,5]. Hence, axonal regeneration is the basis for the restoration of neural function after traumatic, toxic, and inflammatory injury in the peripheral nervous system (PNS), and enhancing regeneration is a potential therapeutic method for improving repair after injury [6,7].

As a neuron matures, the transcriptional program supporting axodendritic formation and extension is dampened, and the neuron becomes a functional unit that specializes in neural transmission. Therefore, robust regrowth of damaged adult axons requires reactivation of the growth pathway in mature neurons, which is often successful after peripheral nerve injury [8,9]. Activation of injury-responsive transcription factors, such as cJun, STAT3, and Smad, is observed in dorsal root ganglion (DRG) neurons after sciatic nerve injury [10,11,12], and many transcriptome analyses have demonstrated that pathways associated with neurite outgrowth are particularly upregulated after injury [13,14,15]. The final outcome of this neuron-intrinsic regeneration program is cytoskeletal regulation at the growth cone that is reformed at the axonal lesion. Dynamic navigation and migration of the growth cone, enforced by actin and microtubule rearrangements, can expedite neural reconnection, which should be achieved before the denervated target tissue undergoes atrophy [16,17,18].

During development and regeneration, the Rho family small GTPases, Rac1, and cell division cycle 42 (CDC42), primarily regulate cytoskeletal dynamics at the growth cone by promoting the polymerization and rearrangement of F-actin and microtubules [19,20]. Rho GTPases are converted from an inactive GDP-bound state to an active GTP-bound state by the action of guanine nucleotide exchange factors (GEF). The responsible GEF can vary depending on the cell type and stimulus [21,22,23]. PAK-interacting exchange factor β (βPix), encoded by the *Arhgef7* gene, is a GEF that can activate Rac1 and CDC42, and previous studies have shown that βPix acts as a positive regulator of cell spreading, neurite development, synapse formation, and social behavior through its GEF activity [24,25,26,27,28,29]. Notably, the alternative splicing of mouse *Arhgef7* RNA resulted in multiple isoforms of βPix, including βPix-a, βPix-b, and βPix-d, with differential expression profiles [30,31,32]. Of these, βPix-b and βPix-d isoforms, which have an insert (INS) region, are specifically expressed in neurons and mainly responsible for the neuronal function of βPix through the regulation of Rac1 activity and microtubule stability [26,30]. However, the role of βPix in axonal regeneration, especially that of the neuronal isoforms, βPix-b and βPix-d, has not been examined yet.

In this study, we investigated the expression of βPix isoforms in peripheral neurons after sciatic nerve injury and tested their requirements for axon regeneration. We found that the neuronal isoforms of βPix were significantly upregulated after nerve injury and that βPix protein localized to the growth cone of regenerating axons. Axon regeneration was attenuated in a mouse model in which the expression of βPix-b and βPix-d isoforms was specifically removed. Inhibiting proto-oncogene c-Src (Src) family kinases, which are known to phosphorylate βPix neuronal isoforms to promote their GEF activity [30], also significantly impairs axonal outgrowth. These data demonstrate that βPix neuronal isoforms regulate peripheral nerve regeneration and support their role as GEF in axon regeneration.

## 2. Results

### 2.1. βPix Protein Levels Increase in the Proximal Stump of the Injured Sciatic Nerve

Expression of many regeneration-associated genes is upregulated by nerve injury. To explore the possibility that βPix is regulated by axonal injury, we examined βPix expression levels after transection of the sciatic nerves (axotomy) (Figure 1A). As the sensory fibers in the sciatic nerve comprise axons of the DRG neurons, mRNA levels of the *Arhgef7* gene, which expresses the βPix protein, were analyzed in axotomized mouse DRG using our previously published whole transcriptome data [33]. At 24 h after axotomy, the *Arhgef7* gene showed a slight but significant upregulation (18.1% increase, *p* = 0.032) (Figure 1B). We investigated 52 additional genes annotated as Rac1 GEFs in the Reactome database (reactome.org) and found that eight genes were significantly upregulated after axotomy. Among the upregulated Rac1 GEFs, the *Arhgef7* gene was the most highly expressed in DRG neuronal cultures, as assessed by RNA sequencing (indicated by bubble color in Figure 1B), supporting its neuronal function.

Among the multiple splice isoforms of the mouse βPix protein, βPix-b and βPix-d isoforms containing the INS region are known to be expressed exclusively in neurons, whereas βPix-a is ubiquitously expressed (Figure 1C) [30]. Therefore, we investigated whether the individual isoforms are differentially regulated in response to nerve injury. We utilized a βPix antibody against the common SH3 domain to detect all βPix isoforms, as well as an anti-INS antibody, to detect the neuronal isoforms βPix-b and βPix-d [26,30]. Western blotting revealed that the βPix-a, βPix-b, and βPix-d isoforms appeared at approximately 75 kD (closed arrowhead), 85 kD (grey arrowhead), and 65 kD (open arrowhead), respectively. In the DRG, where the neuronal cell bodies were located, the three isoforms were detected, but their expression levels did not increase one day after the injury. βPix-d appeared mildly reduced (Figure 1D,E). However, all the isoforms were significantly upregulated in the sciatic nerve stump proximal to the lesion (dotted box in the sciatic nerve, Figure 1A), with isoform βPix-b showing the highest fold induction (2.47-fold, *p* < 0.01) (Figure 1D,E). We further examined the βPix protein levels three days after the nerve injury, when regeneration of injured sciatic nerve axons becomes robust, and we found that there was a sustained increase in the βPix isoform levels in the proximal nerve stump (Figure 1F,G). The βPix levels in the DRG were not significantly altered on day 3, supporting a spatially controlled increase in the βPix levels in injured regenerating axons. Primary DRG neuron cultures showed that βPix-a was hardly detected in neuronal culture lysates, indicating that βPix-b and βPix-d were the major βPix isoforms in DRG neurons (Figure 1H,I).

### 2.2. βPix Protein Is Localized to the Regenerating Axon Tip

To verify the axonal localization of βPix during regeneration, we performed immunofluorescence (IF) staining for βPix in DRG neurons with regrowing axons. A polyclonal antibody raised against the N-terminal region of βPix was used to detect all isoforms. In adult DRG neurons cultured for 20 h, βPix was detected in the growing axons (Figure 2A). In particular, βPix localization was prominent in the lamellipodium area in the peripheral domains of the growth cones (Figure 2A). To directly examine βPix localization in regenerating axons after injury, we performed IF in embryonic DRG spot culture, in which millimeter-long axons can be cut by axotomy with a blade and monitored afterwards [2,35]. Three hours after the axotomy, around which axon regeneration begins, βPix expression was apparent in the regenerating growth cones formed from the cut axons (Figure 2B). To investigate βPix expression during peripheral nerve regeneration, we injured the mouse sciatic nerve by crushing it with fine forceps, after which injured axons regenerated with directionality toward the original target within the maintained epineurium tissue. Consistent with the western blot results of axotomized nerves, βPix expression was increased by the crush injury and was the most prominent in the area distal to the crush where regeneration occurred (Figure 2C). These results demonstrate that βPix protein, which is upregulated after nerve injury, is localized in the regenerating axon shaft and growth cone.

### 2.3. Axon Regeneration Is Impaired in Mice Deficient for the Neuronal Isoforms of βPix

Based on the role of βPix as a GEF for Rac1 and CDC42 GTPases, we tested whether βPix is involved in axon regeneration after injury. We previously generated βPix neuronal isoform knockout (βPix NIKO) mouse line with the genetic removal of exon 19, which causes deletion of the corresponding INS region, leading to the βPix-b and βPix-d isoform-specific deficiency [26]. The βPix NIKO mice are viable, and their reported phenotypes include impairments in neuritogenesis and microtubule stability in cultured hippocampal neurons [26], whereas conventional βPix knockout mice are embryonically lethal [25]. To test the requirement of the neuronal βPix isoforms for regeneration using βPix NIKO mice, we first confirmed the loss of the neuronal isoform expression in the sciatic nerves of βPix NIKO mice (Figure 3A). Adult DRG neurons cultured from the βPix NIKO mice displayed reduced neurite outgrowth (Figure 3B), as assessed by the cumulative distribution of the longest neurite length per neuron, mean neurite length, and percentage of outgrowth failure 16 h after plating (Figure 3C–E). The mean length of the neurites labelled by anti-β3 tubulin, a neuronal marker, was reduced in βPix NIKO cultures by 25.5%, compared to wild-type (WT) cultures. In vivo axon regeneration was also examined after crushing the mouse sciatic nerve with fine forceps, using a regeneration assay based on the regenerating axon-specific marker, SCG10 [2]. In the WT mice, SCG10-positive axons regenerated up to 0.5 mm on day 1 (Appendix A) and 3.5 mm on day 3 after the crush injury (Figure 3F,G), as reported previously [2]. The SCG10-positive axonal area, reflecting axonal regeneration, was not significantly different between WT and βPix NIKO on day 1 after crush injury (Appendix A). However, consistent with the in vitro results, the SCG10-positive axonal area was significantly reduced in the βPix NIKO mice (32.8% decrease) three days after injury, when there was enough dynamic range of regenerated axon length (Figure 3F–H). Collectively, these data demonstrate that βPix neuronal isoforms positively regulate the regeneration of injured axons.

### 2.4. βPix Neuronal Isoforms Are Not Required for a Conditioning Injury Effect

Peripheral axon regeneration is regulated by two distinct mechanisms: conditioning injury effects involving transcriptional regulation by injury and local control of axon extension from the site of injury [36]. Although the two pathways can interact via axon transport of regenerative signals, it is likely that βPix primarily contributes to local axonal growth, considering its GEF activity and axonal localization. To understand the molecular role of βPix for regeneration, we tested if βPix is involved in induction of the conditioning injury effect that is represented by phosphorylation of injury-associated transcription factors, such as cJun and STAT3 [37,38,39], using βPix NIKO mice. We assessed the transcription factor activation 24 h after the nerve crush to minimize potential communication with signals from regrowing axons at a later stage. In the WT mice, nerve crush injury largely induced the phosphorylation of both cJun and STAT3 (Figure 4A). We found that this induction was not significantly altered in the absence of neuronal βPix isoforms (Figure 4B,C). These data support the notion that βPix neuronal isoforms are dispensable for the conditioning injury effect, but they may mainly regulate the extension of injured axons.

### 2.5. Inhibition of Src Kinase Impairs Axonal Outgrowth in DRG Neurons

The successful phosphorylation of cJun and STAT3 in βPix NIKO (Figure 4) suggests that βPix may promote axonal extension via its previously identified function as a GEF that regulates cytoskeletal dynamics through small GTPases. The INS region, unique to neuronal isoforms, contains an Src phosphorylation site (Tyr598 in βPix-b), and we have reported that this phosphorylation promotes Rac1 GEF activity and normal dendritic spine development [30]. To test whether the same pathway is involved in axonal regeneration, we confirmed that phosphorylated Src (Tyr418) was expressed in sciatic nerves, though the levels were not markedly increased by injury (Appendix A). Next, we examined whether axonal regrowth in adult DRG neuronal cultures was impaired by Src inhibition. We treated cultures with a Src kinase inhibitor, PP2 (10 μM), 5 h after plating to rule out its effects on initial cell adhesion and sprouting, and instead focused on possible changes in axon extension [40]. For the same reason, we also grew cultures for 20 h, which was longer than the previous βPix NIKO analysis (Figure 3B). We measured only neurons with neurites to exclude cells that failed to initiate neurites. We stained the cultures for β3 tubulin and measured the longest neurite length per neuron (Figure 5A). Overall, axonal outgrowth was decreased by PP2 treatment, as shown by the higher frequencies of neurons in the short neurite length range in PP2-treated neurons compared to vehicle-treated control neurons (DMSO) (Figure 5B). The mean neurite length in PP2-treated neurons was also significantly decreased compared to the control (38.4% decrease) (Figure 5C). Additionally, we found aberrant growth cones in the PP2-treated neurons, where growth cone morphology and βPix localization were impaired (Figure 5D). In DMSO-treated control cultures, βPix was found in the lamellipodia of fan-shaped growth cones. In the PP2-treated neurons, the growth cones often appear disorganized, and in these cases, βPix localization was mostly restricted to the tubulin-positive area of the growth cones rather than observed in lamellipodia. These results support the requirement of Src activity for the adequate localization of βPix in regenerating growth cones.

Collectively, these results demonstrate that βPix protein expression levels increase significantly in the proximal stump of injured peripheral axons and neuronal isoforms of βPix, βPix-b, and βPix-d, thereby promoting axonal extension during peripheral nerve regeneration, likely through regulation by Src kinase.

## 3. Discussion

Successful regeneration of axons after nerve injury requires the formation of a regenerating growth cone from the tip of the damaged axon and subsequent axon extension. After the axotomy of cultured embryonic DRG neurons, growth cones are reformed at the proximal end of the cut axons within a few hours [37]. Newly developed growth cones can dynamically navigate and migrate, resulting in the extension of millimeter-long axons. Similarly, the formation of growth cones in regenerating axons of the sciatic nerve has been reported using in vivo axon imaging [41]. In contrast, regeneration failure is often associated with the formation of retraction bulbs rather than growth cones. Injured axons in the central nervous system, such as the ascending tract axons of the spinal cord, develop retraction bulbs where microtubules appear looped and disorganized, and eventually degenerate [41,42]. Although the promotion of growth cone formation at the site of injury is critical, the regulators of this process have not been fully identified. In this study, we provided evidence supporting the role of βPix in controlling growth cone formation and axon extension during regeneration. First, βPix protein expression levels were significantly increased after nerve injury in the nerve segment encompassing the regenerating axons. Second, βPix localizes to the growth cone tips. Third, deficiency of the βPix neuronal isoforms leads to delayed axon regeneration. As βPix is a GEF that can activate small GTPases that regulate cytoskeletal rearrangement in the growth cone, the βPix neuronal isoforms likely regulate regeneration by modulating cytoskeletal dynamics through GEF activity. A recent study has shown that TC10 (Rhoq), the Rho family small GTPase that can be activated by βPix [27], is involved in axon regeneration of the mouse hypoglossal and optic nerves [43], raising a hypothesis that βPix may promote regeneration through modulating TC10 function [44]. Future studies are needed to identify the role of TC10 and other small GTPases in peripheral regeneration and to determine whether they are the responsible GTPases in the βPix pathway. 

After nerve injury, the neuronal βPix-b and βPix-d isoforms showed high expression fold changes, and this observation led to the hypothesis that the neuronally expressed isoforms might be mainly responsible for regulating regenerative responses in injured nerves. Prominent gene expression changes are often considered as supportive data for their functional involvement in axonal regeneration. Our group has previously reported a transcriptome analysis in which we defined injury-induced and DLK-dependent gene expression changes [33]. As DLK is a MAP3K required for delivering JNK-mediated local injury signals to the cell body, which in turn enhances axon regeneration, DLK-dependent gene expression changes have been suggested as a criterion for selecting potential regulators of regeneration [37]. Similarly, Tedeschi et al. investigated injury-induced genes and their regulation during neuronal maturation based on the notion that genes important in axonal elongation may show decreased expression levels as neurons mature [9]. From our analysis of potential Rac1 GEF genes that were upregulated after the injury, we also identified the *Gna13* gene. The *Gna13* gene is known as a GEF for RhoA, and its activation results in growth cone collapse, which contradicts the hypothesis that Gna13 positively regulates axon regeneration [45]. However, Gna13 can also activate AP-1, a key injury-responsive transcription factor complex, so it might play a role independent on the RhoA GEF function [46]. Gna13 is highly expressed in injured mouse DRG and cultured DRG neurons, further supporting its potential involvement in regeneration.

Compared with the RNA-seq results showing only a slight induction of *Arhgef7* mRNA expression in DRG by nerve injury, protein analysis showed large increases in βPix isoforms. Additionally, the βPix protein levels increased only in the sciatic nerve segment but not significantly in the DRG tissues. These discrepancies suggest that post-transcriptional regulation is crucial for controlling the protein levels. For example, *Arhgef7* mRNA may be transported to injured axon terminals and locally translated. Alternatively, βPix proteins newly synthesized in the cell body may be rapidly transported to the axon terminal. An altered protein turnover rate may also contribute to the accumulation of βPix protein in injured nerve segments. Since βPix-b and βPix-d are the major isoforms in neurons, βPix-a is likely to be expressed mostly in non-neuronal cells in the tissues, including Schwann cells, satellite glia, macrophages, and fibroblasts. It would be interesting to investigate these possibilities to determine the mechanisms underlying the regulation of each βPix isoform.

Src kinase can phosphorylate a variety of substrates, including many signaling molecules involved in tumorigenesis, when activated by upstream signals, such as receptor tyrosine kinases like EGFR [47]. With respect to mechanotransduction, Src can be activated by integrins, and it plays a pivotal role in promoting cell adhesion, polarization, and migration, as well as axon outgrowth and dorsal root regeneration [48], by mediating phosphorylation of FAK, paxillin, PAK, and Rho-GEFs at their tyrosine residues [49,50]. In our previous study, we identified that Src phosphorylates βPix-b at tyrosine 598 within the INS region, which is specific to neuronal βPix isoforms, during dendritic spine formation and synaptogenesis. Src-mediated phosphorylation of βPix-b enhances its Rac-GEF activity and is associated with spine development [30]. Additionally, a previous study by Zhao et al. demonstrated that active Src is upregulated after nerve injury and expressed in regenerating axons and Schwann cells of crushed rat sciatic nerve, supporting a potential role of Src during peripheral nerve regeneration [51]. In line with these previous findings, the current study found that active Src is present in sciatic nerves and that Src activity is required for neurite reformation in DRG neurons. Our data suggest that Src is necessary for normal growth cone morphogenesis and the localization of βPix to the peripheral domain of the growth cone. Whether Src acts through the phosphorylation of the INS region in the axon regeneration pathway needs to be investigated in future studies.

Conventional knockout of βPix in mice is embryonically lethal, and severe defects, such as failure of allantois-chorion fusion, are observed in affected embryos at embryonic day 9.5 [25]. Therefore, the βPix NIKO mouse line, in which the neuronal isoform expression is replaced with ubiquitous βPix-a expression (and possibly INS-deleted βPix-d), is a valuable model for studying the specific roles of neuronal isoforms within neurons. However, a potential drawback of this genetic reagent is that βPix-a isoform is expressed at a higher level than in WT animals [26] and may affect phenotypes. To overcome this problem, feasible approaches would include designing isoform-specific knockdown methods using adeno-associated viral delivery to DRG neurons.

Sciatic nerve injury is a well-established experimental model where regeneration of long peripheral axons, injury-induced retrograde signaling, and functional outcomes of nerve repair can be examined. However, studies using various regeneration models, in other mammalian peripheral nerves or other model organisms, e.g., zebrafish and *Drosophila*, indicate that the underlying molecular mechanisms and the success of regeneration vary in different models [52]. Moreover, axonal regeneration in the central nervous system is extremely limited—after spinal cord injury, both dorsal column axons and cerebrospinal tract axons fail to regenerate beyond the lesion site. Hence, the role of βPix is likely to be different between the peripheral and central axon regeneration and needs further investigation to elucidate its requirement and the underlying molecular mechanisms. During differentiation of mouse hippocampal neurons, we previously showed that βPix interacts with PAK, which in turn phosphorylates Stathmin-1 to induce microtubule stability [26]. The small GTPases that can be activated by βPix, CDC42, and TC10, are known to interact with WASP family proteins to initiate actin polymerization through regulation of the Arp2/3 complex [53]. Understanding precise mechanisms underlying the βPix’s function during regeneration will require identification of the downstream pathways regulating cytoskeletal dynamics among the aforementioned mechanisms, as well as other interacting signaling pathways, in each injury paradigm. As small GTPases and their GEFs are also associated with other cellular events that proceed with tissue repair, such as cell proliferation, migration, and chemotaxis, interaction with these pathways will also need to be considered [54]. Finding the precise mechanisms will also help to assess the therapeutic opportunities targeting βPix as a method to improve axon regeneration.

Importantly, βPix isoforms are highly conserved between rodents and humans; therefore, our finding on the specific role of βPix is likely to be well conserved in patients with nerve damage. Axon regeneration is implicated in recovery after traumatic injury and may underlie the pathogenesis of many neuropathic conditions. Therefore, modulating the action of βPix and associated function of Src kinase, as well as targeting small GTPases, may provide a good therapeutic opportunity to improve recovery from neural damage.

## 4. Materials and Methods

### 4.1. Antibodies and Reagents

Rabbit anti-β3 tubulin antibodies were custom-generated by Abclon (Seoul, Korea), using N-MYEDDDEESEAQGPK-C as a peptide antigen. The βPix antisera against the SH3 domain and INS region have been described previously [30]. The following antibodies were purchased: rabbit anti-ARHGEF7 (βPix) (Novus Biologicals, Centennial, CO, USA, NBP2-92602), rat anti-α tubulin (Santa Cruz, Dallas, TX, USA, sc-53030), rabbit anti-SCG10 (Novus Biologicals, Centennial, CO, USA, NBP1-49461), rabbit anti-p-cJun (Cell Signaling, Danvers, MA, USA, CST9261), rabbit anti-p-STAT3 Y705 (Cell Signaling, Danvers, MA, USA, CST9145), rabbit anti-STAT3 (Santa Cruz, Dallas, TX, USA, C-20), mouse anti-β3 tubulin (BioLegend, San Diego, CA, USA, 801202), chicken anti-β3 tubulin (Abcam, Waltham, MA, USA, ab41489 ), rabbit anti-p-Src Y418 (Thermo Fisher, Waltham, MA, USA, 44-660G), rabbit anti-cJun (Cell Signaling, Danvers, MA, USA, CST9165), Cy3-conjugated anti-mouse secondary antibody (Sigma, St. Louis, MO, USA, AP192C), Cy3-conjugated anti-rabbit secondary antibody (Sigma, St. Louis, MO, USA, AP182C), Alexa Fluor 488-conjugated anti-mouse secondary antibody (Invitrogen, Waltham, MA, USA, R37114), Alexa Fluor 488-conjugated anti-rabbit secondary antibody (Invitrogen, Waltham, MA, USA, A-21206), and Alexa Fluor 594-conjugated anti-chicken secondary antibody (Abcam, Waltham, MA, USA, ab150176). To selectively inhibit Src tyrosine kinases, PP2 (Tocris, Bristol, UK, 1407) was used.

### 4.2. Mice and Surgery

We used 6–8-week-old C57BL/6 mice purchased from Orient Bio Inc. (Seongnam, Korea) or βPix WT and NIKO mice maintained in an animal facility at Dong-A University. βPix NIKO genotypes were confirmed by tail genotyping polymerase chain reaction, using the following primer pairs: forward primer 5′-AGC ACA GTT GAC GTT GCT TTC TGT C-3′, WT specific reverse primer 5′- AAA GCC CAT CAG GTA CTC ACT GGA C-3′ and KO specific reverse primer 5′- AAA CTA TCA GTC TGC CCT CAC CCA C-3’. Mouse husbandry and surgical procedures were conducted following the animal protocol approved by the Dong-A University Committee on Animal Research under the guidelines established by the Korean Academy of Medical Sciences. Female and male mice were used in the experiments. For nerve injury, the mice were anesthetized using isoflurane (Piramal, Mumbai, India) inhalation, and the sciatic nerve was exposed by a small incision on the skin and muscle and transected using surgical scissors (axotomy). Crush injury was given for 5–10 s by using fine forceps (Fine Science Tools, Foster City, CA, USA, Dumont #55). After sciatic nerve injury, the incision was closed using a nylon 6-0 suture (AILEE, Busan, Korea, NK621). At 1–3 days after injury, the mice were euthanized, and tissue samples were dissected for analysis.

### 4.3. Primary DRG Neuron Cultures

Cultures were prepared from DRG, as previously described [37]. Briefly, adult DRG at L4–5 were dissected from the mice and incubated in DMEM (Gibco, Waltham, MA, USA, 11965-092)/Liberase TM (Roche, Basel, Switzerland, 5401119001)/DNase I (Sigma, St. Louis, MO, USA, DN25)/1% bovine serum albumin and in 0.05% trypsin-EDTA (Gibco, Waltham, MA, USA, 25300-054), each for 15 min. The tissue was then dissociated and plated in DMEM supplemented with 10% fetal bovine serum (FBS), 1% Glutamax (ThermoFisher, Waltham, MA, USA, 35050061), and 1% penicillin-streptomycin (ThermoFisher, Waltham, MA, USA, 15140163) on Lab-Tek chambers (ThermoFisher, Waltham, MA, USA, 177437) coated with 0.1 mg/mL poly-D-lysine (Sigma, St. Louis, MO, USA, P0899) and 10 μg/mL laminin (Invitrogen, Waltham, MA, USA, 23017-015). For embryonic DRG cultures, DRG tissues collected from embryonic day 13.5 mice were triturated after 22 min incubation in 0.05% trypsin-EDTA. Dissociated cells were plated in a Neurobasal medium (Gibco, Waltham, MA, USA, 21103-049) supplemented with 2% B-27 (Gibco, Waltham, MA, USA, 17504-044), 1% Glutamax, 1% penicillin-streptomycin, 1 μM 5-fluoro-2′-deoxyuridine (Sigma, St. Louis, MO, USA, F0503), 1 μM uridine (Sigma, St. Louis, MO, USA, U3003), and 50 ng/mL 2.5S nerve growth factor (Envigo, Indianapolis, IN, USA, BT-5017) on Lab-Tek chambers coated with 0.1 mg/mL poly-D-lysine and 3 μg/mL laminin. For the axotomy, the axons were manually cut with a flat blade (Fine Science Tools, Foster City, CA, USA, 10035-10/10035-12) at days in vitro (DIV) 7. To estimate the neuronal expression of the Rac1 GEF genes, DIV7 embryonic DRG cultures were subjected to RNA sequencing using a direct RNA sequencing kit (Oxford Nanopore Technologies, Oxford, UK, SQK-RNA002).

### 4.4. In Vitro Axon Outgrowth Assay and IF

In vitro, axon outgrowth assay was performed by analyzing the neurite length in adult DRG neurons, as previously reported [37,40]. Briefly, cultured adult DRG cells were fixed with 4% paraformaldehyde (BioSolution, Seoul, Korea, BP031a) for 16–20 h after plating. Fixed cells were followed by blocking with 5% FBS/3% bovine serum albumin in phosphate-buffered saline/0.1% TritonX-100 (PBS-T), staining with anti-β3 tubulin rabbit antibody for an axonal marker, and mounting with DAPI-containing mounting medium (Vector Laboratories, Newark, CA, USA, H-1200). At least 50 cells per sample group were imaged using an EVOS M7000 imaging system (Thermo Fisher Scientific, Waltham, MA, USA, AMF7000) or a charge-coupled camera device (DS-Qi2) on a fluorescence microscope (Ti-E, Nikon, Tokyo, Japan) with a 10× or 20× objective lens. The longest neurite per neuron was measured using ImageJ (National Institutes of Health (NIH), Bethesda, MD, USA) with the NeuronJ plug-in [55]. For βPix localization studies, fixed cells were immunostained with rabbit anti-ARHGEF7 antibody (Novus Biologicals, Centennial, CO, USA,) and mouse anti-β3 tubulin antibody, and the representative micrographs were acquired using a Zeiss Imager M2 in ApoTome II microscope (Carl Zeiss, Oberkochen, Germany) equipped with a 40× water immersion lens at the Neuroscience Translational Research Solution Center (Busan, Korea).

### 4.5. In Vivo Axon Regeneration Assay and IF

The sciatic nerves were dissected three days after the injury. The tissues were fixed in 4% paraformaldehyde for 1.5 h and then immersed in 30% sucrose in PBS at 4 °C until they were cryopreserved in an OCT medium (Tissue-Tek, Tokyo, Japan, 4583). Samples were then longitudinally cryosectioned at 10 μm thickness. Cryosections were stained with rabbit anti-SCG10 antibody and chicken anti-β3 tubulin antibody in a blocking solution, 5% FBS/3% BSA in PBS-T (0.1% TritonX-100). Immunofluorescence images were acquired using an EVOS M7000 imaging system (Thermo Fisher Scientific, Waltham, MA, USA, AMF7000) or a DS-Qi2 camera on a fluorescence microscope (Ti-E, Nikon, Tokyo, Japan) with a 10× or 20× objective lens. To assess axon regeneration, SCG10 staining intensity was measured from the proximal to distal directionality with binning by 50 pixels in width and expressed as the relative intensity to the crush site, which was defined as the point with the maximal SCG10 intensity, as reported previously [37].

### 4.6. Immunoblotting

Sciatic nerves or DRG were homogenized in 1× lysis buffer (Cell Signaling, Danvers, MA, USA, 9803), containing protease inhibitor (Merck, St. Louis, MO, USA, 4693159001) and phosphatase inhibitor (Merck, St. Louis, MO, USA, 4906845001) cocktails. Tissue homogenates were subjected to centrifugation at 12,000× *g* for 10 min at 4 °C, and the resulting supernatants were then subjected to protein assay using a DC protein assay kit (Bio-Rad, Hercules, CA, USA, 5000116). Samples resolved by SDS polyacrylamide gel electrophoresis and transferred to a nitrocellulose membrane were subjected to immunoblotting for p-cJun, p-STAT3, STAT3, βPix, and α tubulin following a standard procedure described previously [37]. Enhanced chemiluminescence was detected using SuperSignal™ West Dura Substrate (Thermo Fisher, Waltham, MA, USA, 34075) with a LAS 4000 imager (GE Healthcare, Chicago, IL, USA) and quantified using ImageJ (NIH, Bethesda, MD, USA).

### 4.7. Statistics

Statistical significance was tested using GraphPad Prism Software 9.50 (San Diego, CA, USA). *p*-values were obtained using Student’s two-tailed *t*-test, and the data in graphs were expressed as mean ± standard error of the mean. All the western blot results were quantified using at least three biological replicates.

## Figures and Tables

**Figure 1 ijms-24-14357-f001:**
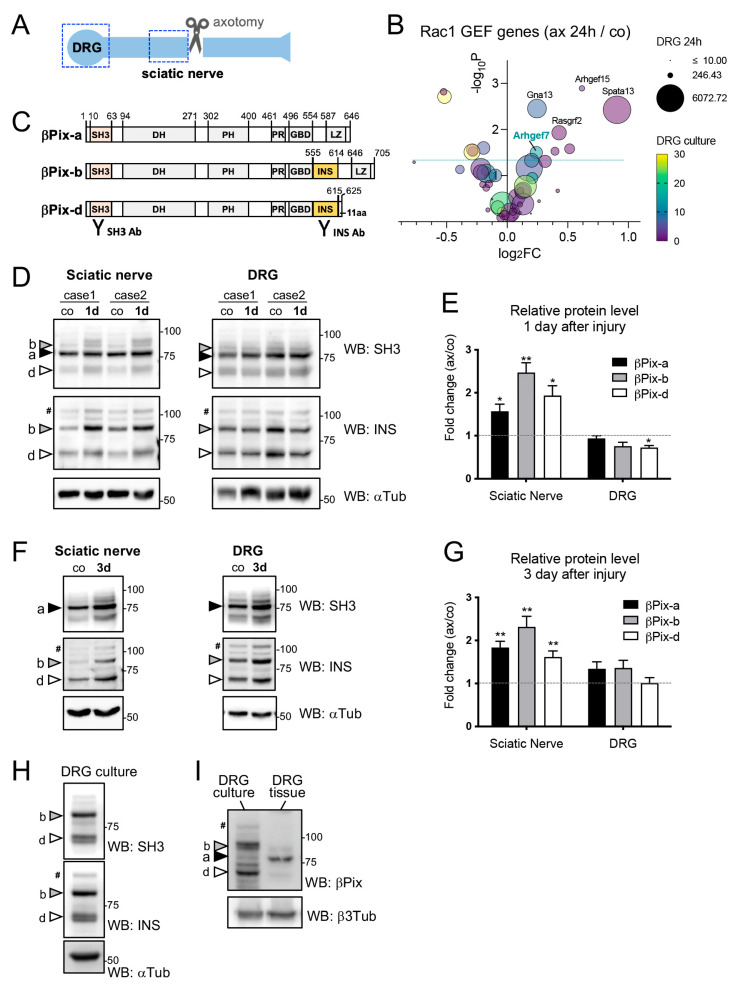
βPix expression increases in the proximal nerve stump after sciatic nerve transection. (**A**). Schematic illustration of peripheral nerve injury. For axotomy, the sciatic nerve was transected at the mid-thigh level. (**B**). RNA sequencing results showing the expression of 53 Rac1 guanine-nucleotide exchange factor (GEF) genes. The volcano scatter plot displaying expression fold change (log_2_FC) of each gene between control DRG and injured DRG (sciatic nerve axotomy, 24 h). The symbol size and color represent average gene counts in injured DRG and cultured DRG neurons, respectively. The dotted line corresponds to *p* = 0.05. (**C**). Schematic diagram of the protein domains of βPix isoforms, βPix-a, βPix-b, and βPix-d. INS region-containing isoforms, βPix-b and βPix-d, are expressed specifically in neurons. Anti-SH3 and anti-INS antibodies are used to detect all isoforms and the neuronal isoforms, respectively. The numbers denote amino acid positions. (**D**). βPix levels are examined by immunoblot analyses of the lysates from the sciatic nerve and DRG at day 1 after axotomy using anti-SH3 and anti-INS antibodies. α tubulin is used as a loading control. (**E**). Quantification of the results shown in (**D**). The βPix levels were normalized with the loading control and plotted as fold changes by injury in individual isoforms. In the sciatic nerve, expression of all βPix isoforms significantly increase at day 1 after the axotomy, with the highest fold induction of βPix-b. βPix levels in the DRG do not increase after injury, but show a decrease in βPix-d levels (27.7% decrease). (**F**). Immunoblot analyses of the lysates from the sciatic nerve and DRG at day 3 after axotomy using anti-SH3 and anti-INS antibodies. α tubulin is used as a loading control. (**G**). Quantification of the results shown in (**F**). βPix levels in the sciatic nerves are significantly increased at day 3 after injury, whereas βPix levels in the DRG are not significantly altered. (**H**). βPix-b and βPix-d protein levels in pure-neuronal embryonic DRG cultures. Immunoblotting with anti-SH3 antibody demonstrates that expression of the neuronal βPix isoforms, βPix-b and βPix-d, is dominant in the DRG neurons, whereas βPix-a isoform is rarely expressed. α tubulin is used as a loading control. (**I**). Lysates from pure-neuronal embryonic DRG cultures and mouse DRG tissues were analyzed for a side-by-side comparison. β3 tubulin is used as a loading control. DRG, dorsal root ganglion; ax, axotomized; co, control; SH3, Src homology 3 domain; DH, Dbl homology domain; PH, Pleckstrin homology domain; PR, proline-rich domain, GBD, Git1-binding domain; INS, insert region; LZ, leucine zipper domain. Black arrowhead, βPix-a; gray arrowhead, βPix-b; white arrowhead, βPix-d; #, βPix-b_L_ isoform generated by an additional splicing event not described in (**C**) [34]. * *p* < 0.05, ** *p* < 0.01 by *t* test; mean ± standard error of the mean (SEM).

**Figure 2 ijms-24-14357-f002:**
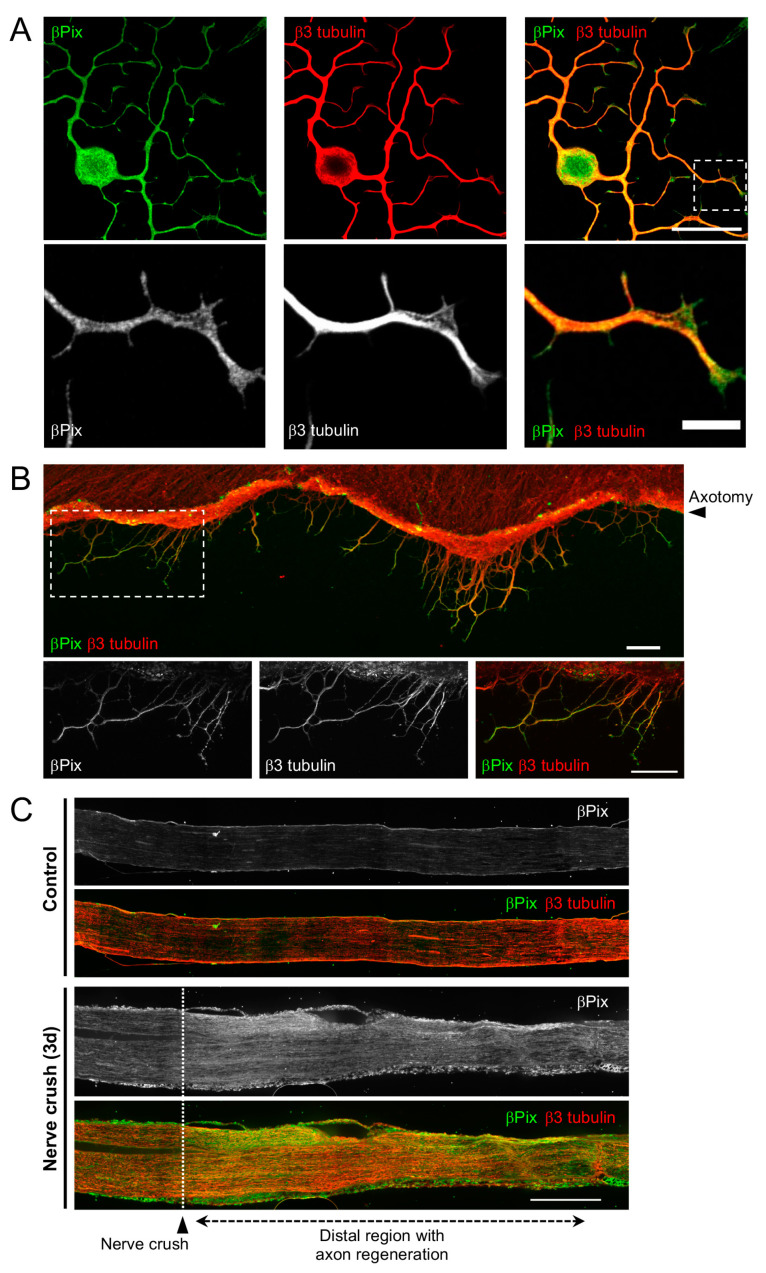
βPix localizes to regenerating axon terminal. (**A**). Dorsal root ganglion (DRG) neurons cultured from adult C57BL/6 mice were incubated for 20 h and immunostained with anti-βPix and anti-β3 tubulin (a neuronal marker) antibodies. βPix localizes in the growth cone tip. Scale bar, 50 μm in upper panels and 10 μm in lower panels. (**B**). Embryonic mouse DRG “spot cultures” were axotomized at days in vitro 7 and allowed for axon regeneration for 3 h. Representative immunostaining images showed that βPix is highly expressed at the tip of regenerating axons. The area in the dotted box in the top row is magnified in the bottom row. Scale bar, 50 μm. (**C**). Longitudinal cryosections of control and crushed mouse sciatic nerves immunostained with anti-βPix and anti-β3 tubulin antibodies. The crushed nerves were collected three days after the surgery to observe the regenerating axons after the crush injury. βPix protein levels increased after injury in the distal sciatic nerve region with regenerating axons. Scale bar, 500 μm.

**Figure 3 ijms-24-14357-f003:**
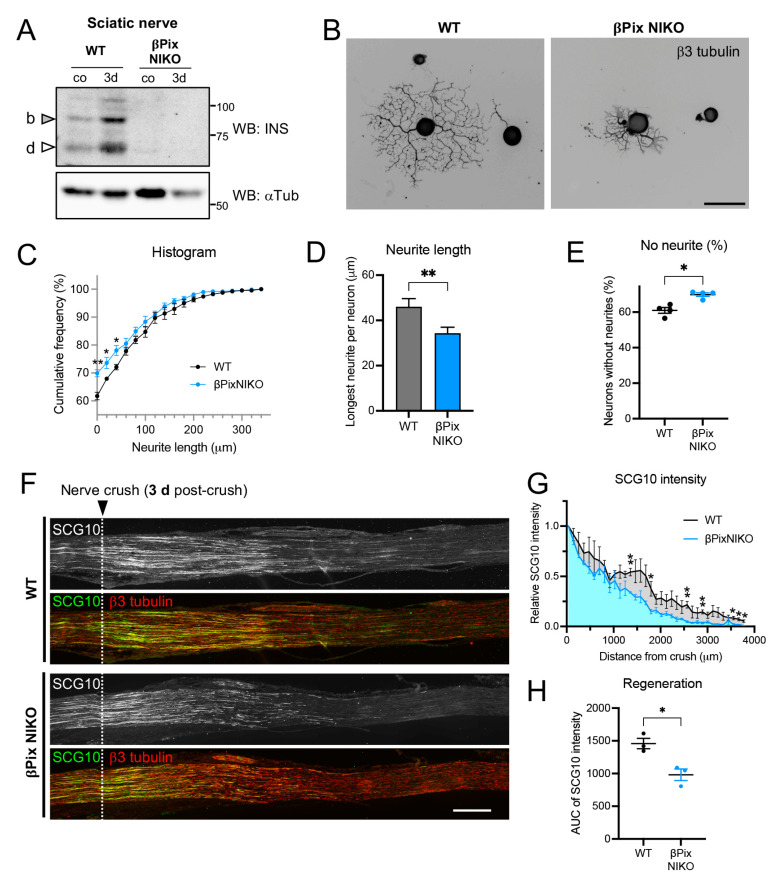
βPix neuronal isoforms regulate axon regeneration. (**A**). Expression of βPix neuronal isoforms, βPix-b and βPix-d, was removed in the sciatic nerve of βPix neuronal isoform knockout (NIKO) mouse. *α*Tub, loading control; co, uninjured; 3d, three days after axotomy. (**B**). Representative immunostaining images of adult DRG neurons cultured from WT and βPix NIKO mice. The neurons were fixed at 16 h after plating and stained for β3 tubulin. Fluorescence images were converted to grayscale for clear visualization of neurites. Scale bar, 100 μm. (**C**–**E**). Quantitative analysis of the results shown in (**B**). (**C**), Cumulative frequency plot of the length of the longest regenerating neurite per DRG neuron. (**D**), Mean neurite length is calculated from the lengths of the longest neurite per neuron. (**E**), Percentage of neurons without neurites. A neurite is defined as a neuronal process longer than the cell body diameter. Data are acquired from four independent experiments, *n* ≥ 79 from each experiment. (**F**). Longitudinal cryosections of control and crushed mouse sciatic nerves immunostained with anti-β3 tubulin and anti-SCG10 antibodies three days after crush injury. Scale bar, 500 μm. (**G**). Data shown in (**F**) were quantified for SCG10 intensity relative to the intensity at the injured site (*n* = 3). (**H**). Area under curve (AUC) of SCG10 intensity plot shown in (**G**) were compared between WT and βPix NIKO mice. Axon regeneration presented by the AUC values is significantly impaired in the βPix NIKO mice (*n* = 3). * *p* < 0.05, ** *p* < 0.01 by *t* test; mean ± SEM.

**Figure 4 ijms-24-14357-f004:**
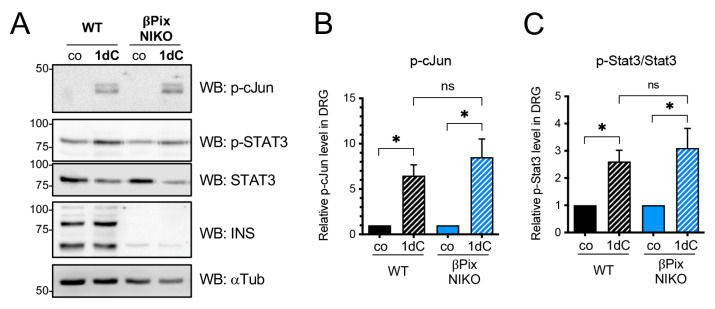
βPix neuronal isoforms are not required for the conditioning injury effect. (**A**). Immunoblot analysis of the DRG lysates collected one day after unilateral sciatic nerve crush injury. co, uninjured nerve; 1dC, crushed nerve at day one after the crush injury. Injury-induced increases in p-cJun and p-Stat3 levels are not significantly altered in the absence of βPix-b and βPix-d isoforms. α tubulin is used as a loading control. (**B**,**C**). Quantification of p-cJun (**B**) and p-Stat3/Stat3 (**C**) shown in (**A**). * *p* < 0.05; ns, not significant by *t* test; mean ± SEM.

**Figure 5 ijms-24-14357-f005:**
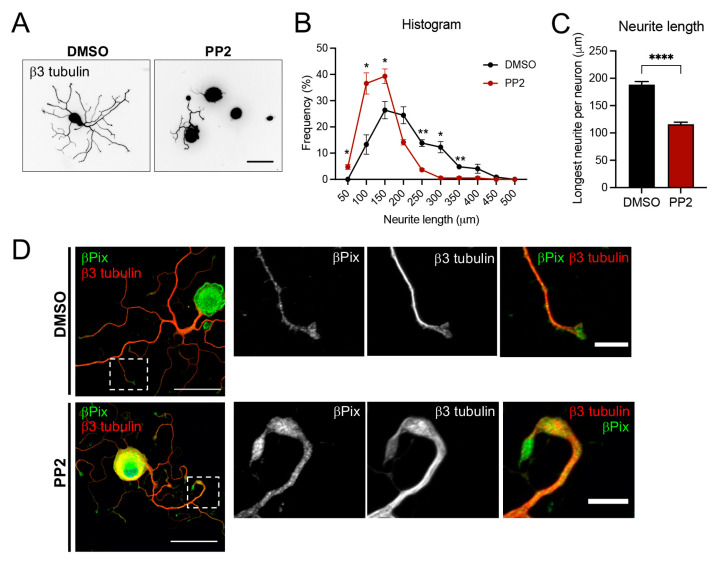
Src inhibition impairs neurite growth in cultured DRG neurons. (**A**). Representative images of WT adult DRG neurons treated with DMSO (vehicle) or Src inhibitor (PP2). The neurons were fixed at 20 h after plating and stained with anti-β3 tubulin antibody. Scale bar, 100 μm. (**B**,**C**). Quantification of the longest neurite length per neuron from the data shown in (**A**). Note that neurons without neurite outgrowth were excluded to rule out the effect of PP2 on cell adhesion. Frequency distribution histogram (**B**) and mean neurite outgrowth (**C**). The longest neurite length per neuron is significantly reduced by Src inhibition. Data are acquired from three independent experiments, *n* ≥ 63 from each experimental group. (**D**). Immunofluorescence images showing βPix protein localization and growth cone morphology in control and PP2-treated DRG neuron cultures. The area in the dotted box in the left column is magnified. Scale bar, 50 μm in left column and 10 μm in right column. * *p* < 0.05, ** *p* < 0.01, **** *p* < 0.0001; mean ± SEM.

## Data Availability

The data presented in this study are available in Appendix A.

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
