# Peer review of "βPix Guanine Nucleotide Exchange Factor Regulates Regeneration of Injured Peripheral Axons"

_ijms, 2023, doi:10.3390/ijms241814357_

Round 1
Reviewer 1 Report
The manuscript by Jeon, Y. et al. about the role of betaPix in the regulation of axonal regeneration appears well-organized, with beautiful iconography and quite interesting. Authors performed a nice experiment, in my opinion.
The only issue I am able to mention is that an abbreviation list could be very helpful sometimes, as the text has several acronyms.
The text also has a generally good grammar / style.
Author Response
We are grateful for the opportunity to present a revised version of our manuscript “βPix Guanine Nucleotide Exchange Factor Regulates Regeneration of Injured Peripheral Axons”. The reviewers were enthusiastic about our manuscript and had constructive suggestions for improvements. We have addressed all of the comments of the reviewers, by revising Fig 1 and Fig 3, adding new supplementary figures (Fig S1 and S2), and extending our discussion to cover the issues raised by the reviewers. Fig 1 now has a new panel, Fig 1I, that shows a side-by-side comparison of the βPix isoforms in DRG neurons versus tissue. In Fig 3A, we included the injured βPix KO lane as requested. Fig S1 shows axonal regeneration one day after crush injury to monitor the time course of regeneration. In Fig S2, we show the p-Src levels in the sciatic nerve before and after injury. We believe that the revised manuscript is improved and hope that you and the reviewers will now find it to be appropriate for publication. A detailed response to individual reviewers is provided below (responses are in blue) and the manuscript was revised accordingly.
Reviewer 1
The reviewer found the paper to be “well-organized, with beautiful iconography and quite interesting” and based on “a nice experiment”. The reviewer had one suggestion:
1) The only issue I am able to mention is that an abbreviation list could be very helpful sometimes, as the text has several acronyms.
- We appreciate this suggestion and present the requested list as a glossary/abbreviation table in the back matter of the manuscript (page 15). The table title is highlighted in yellow in the text.
Reviewer 2 Report
This article describes a study that identifies βPix as a regulator of axonal regeneration. The study shows that βPix promotes neurite regrowth in cultured dorsal root ganglion neurons and in vivo axon regeneration after sciatic nerve crush injury. The study also demonstrates that βPix is localized in the peripheral domain of the growth cone in regrowing axons, suggesting that it may play a key role in growth cone motility and axon guidance.
1. Some potential limitations of the study discussed in the paper could include:
1-1. The focus is on the role of βPix in axonal regeneration and not addressing other potential factors that may contribute to this process.
1-2. The study is limited to specific experimental conditions and may not be generalizable to other contexts or conditions.
1-3. The study does not provide a detailed mechanistic understanding of how βPix regulates axonal regeneration, and further research is needed to elucidate the underlying mechanisms fully.
2. Based on the findings discussed in this paper, I would like to suggest some future studies:
2-1. Investigating the role of βPix in other contexts, such as nerve injuries and the central nervous system, is warranted.
2-2. The potential of βPix as a therapeutic target to promote axonal regeneration and functional recovery after nerve injury warrants investigation.
2-3. The mechanisms that βPix regulates axon regeneration, including interactions with other signaling pathways and cytoskeletal components, require further elucidation.
2-4. Investigation of the role of TC10 and other small GTPases in axonal regeneration and identification of potential targets for drug development are needed.
In addition to the above suggestions, I would like to pose a specific question:
3. Please describe the difference between the outgrowth and regrowth of axons at the gene level and cell signaling level in the Discussion.
4. Figure 2C and Figure 3F show data from 3 days after nerve crush, so we don't know if regeneration of axons actually occurred. The correct data should show that regeneration of axons occurs gradually from 0 hours to 1, 2, and 3 days after nerve crush. Perhaps the data should be supplemented further. Can you show this be shown as supplementary data if needed?
I think this is plagiarism. Please rewrite.
line 23: Using a genetic mouse model in which....
line 35: Axons in the peripheral nervous system (PNS) are ......
line 61: ....activated by guanine nucleotide exchange factors (GEF), which....
line 84: Peripheral nerve injury induces changes in the expression....
line 290: ...axons in the dorsal column of the spinal cord....
line 308: The correlation between changes in gene expression and.......
line 447: Tissue homogenates were centrifuged at 12,000 × g for .....
line 448: …4oC, and the protein concentrations in the...
lines 449-450: Equal amounts of protein were resolved by SDS polyacrylamide gel electrophoresis, transferred to a nitrocellulose membrane, and ......
line 456: Statistical analyses were performed using GraphPad Prism software.....
Minor editing of English language required
Author Response
We are grateful for the opportunity to present a revised version of our manuscript “βPix Guanine Nucleotide Exchange Factor Regulates Regeneration of Injured Peripheral Axons”. The reviewers were enthusiastic about our manuscript and had constructive suggestions for improvements. We have addressed all of the comments of the reviewers, by revising Fig 1 and Fig 3, adding new supplementary figures (Fig S1 and S2), and extending our discussion to cover the issues raised by the reviewers. Fig 1 now has a new panel, Fig 1I, that shows a side-by-side comparison of the βPix isoforms in DRG neurons versus tissue. In Fig 3A, we included the injured βPix KO lane as requested. Fig S1 shows axonal regeneration one day after crush injury to monitor the time course of regeneration. In Fig S2, we show the p-Src levels in the sciatic nerve before and after injury. We believe that the revised manuscript is improved and hope that you and the reviewers will now find it to be appropriate for publication. A detailed response to individual reviewers is provided below (responses are in blue) and the manuscript was revised accordingly.
Reviewer 2
The reviewer stated that there were “some potential limitations of the study” and asked us to address several issues. A point-by-point response is provided below.
1) Some potential limitations of the study discussed in the paper could include:
1-1) The focus is on the role of βPix in axonal regeneration and not addressing other potential factors that may contribute to this process.
- We appreciate the comment and agree with the reviewer that our study is focused on the role of βPix in the regulation of axonal regeneration in a limited experimental setting. Although we could not substantially expand the scope of our study during the revision process, we revised our manuscript to discuss the limitations of the current study and the future opportunities. The revised sentences are highlighted in yellow. We have addressed other factors that may contribute to the regeneration processes such as cell migration and proliferation in the discussion section (lines 422-426).
1-2) The study is limited to specific experimental conditions and may not be generalizable to other contexts or conditions.
- As also addressed in the reviewer’s concern 2-1), we now discuss other experimental paradigms that may help to extend our understanding of βPix’s role in regeneration, such as spinal cord injury (lines 404-414). We agree that these other experimental paradigms, especially those in the central nervous system, are worthwhile to investigate and therefore suggest them as a direction for future studies.
1-3) The study does not provide a detailed mechanistic understanding of how βPix regulates axonal regeneration, and further research is needed to elucidate the underlying mechanisms fully.
- We agree with the reviewer that our mechanistic understanding is still limited. Our study can be extended to elucidate the downstream pathway of βPix, by identifying the involved GTPases, testing the requirement of the known βPix-interacting kinase, PAK, and determining the effector molecules such as regulators of cytoskeletal dynamics. Although we could not approach these issues experimentally, we revised the manuscript to address them in the discussion (lines 414-421).
2) Based on the findings discussed in this paper, I would like to suggest some future studies:
- We appreciate the reviewer’s suggestion for important future directions. Though we were not able to carry out the suggested studies during the revision process due to the time limit, we addressed the issues in the discussion section as indicated below.
2-1) Investigating the role of βPix in other contexts, such as nerve injuries and the central nervous system, is warranted.
- We used dorsal root ganglion neuronal cultures and sciatic nerve injury experiments which are the most-used models in peripheral nerve injury studies. Because the central nervous system (CNS) regeneration capacity and the underlying mechanisms are quite different from those in the peripheral nervous system (PNS) injury, the role of βPix may also be different in CNS injury paradigms. We have now added discussion of potential βPix functions in additional injury models in the text (lines 404-414).
2-2) The potential of βPix as a therapeutic target to promote axonal regeneration and functional recovery after nerve injury warrants investigation.
- We revised the manuscript to discuss the novel therapeutic opportunities targeting the βPix pathway in lines 424-426 and 431-432.
2-3) The mechanisms that βPix regulates axon regeneration, including interactions with other signaling pathways and cytoskeletal components, require further elucidation.
- We discussed the future study direction involving the dissection of the signaling pathway through which βPix regulates axonal regeneration in lines 414-421.
2-4) Investigation of the role of TC10 and other small GTPases in axonal regeneration and identification of potential targets for drug development are needed.
- We now provide discussion of the future directions regarding 1) the role of the small GTPases and 2) future drug development targeting the GTPases in lines 341-343 and 431-432.
3) Please describe the difference between the outgrowth and regrowth of axons at the gene level and cell signaling level in the Discussion.
- The terms outgrowth and regrowth are often used interchangeably to indicate 1) regeneration of severed axons and 2) outgrowth of newly formed neurites in cultured adult neurons, which reflects the neuronal regeneration capacity. While regrowth is specifically used to describe the regenerative growth of injured axons, outgrowth can be widely used for axonal extension during development. In the original manuscript, we used the two terms interchangeably, but the reviewer’s comment made us realize that it could mislead readers. To avoid any confusion, we have revised the manuscript to consistently use “regrowth” to refer to regenerative outgrowth. However, the in vitro assay for DRG neurite length is more commonly called a “neurite outgrowth assay”, so we specifically used the term “outgrowth” when describing results obtained from the assay (lines 31, 34, 87, 207, 208, 270, 488, 489).
4) Figure 2C and Figure 3F show data from 3 days after nerve crush, so we don't know if regeneration of axons actually occurred. The correct data should show that regeneration of axons occurs gradually from 0 hours to 1, 2, and 3 days after nerve crush. Perhaps the data should be supplemented further. Can you show this be shown as supplementary data if needed?
-We appreciate the comment. The time-course of axon regeneration as labeled by SCG10 immunostaining was reported in detail in our previous publication (Shin et al. Experimental Neurology, 2014, reference # [2]). In the revised manuscript, we have added a new supplementary figure (Figure S1) to help readers understand that regeneration of crushed axons occurs gradually. We show anti-SCG10 immunofluorescence images from an uninjured WT nerve (Figure S1A) and 1 day-post-injury (DPI) nerves from WT and βPix NIKO mice (Figure S1B). Results from axon regeneration assay at 1 DPI are also provided in Figure S1C-S1D, but regeneration in βPix NIKO was not significantly different from that in WT at this time-point. We speculate that the initial regeneration speed at 1 DPI is too slow, compared to that in the later stage (e.g. day 3), to allow for a sufficient dynamic range in the regeneration length. The result is described in lines 214-217 of the revised manuscript and highlighted in yellow.
5) I think this is plagiarism. Please rewrite.
line 23: Using a genetic mouse model in which.... : line 29 in revision
line 35: Axons in the peripheral nervous system (PNS) are ...... : line 41 in revision
line 61: ....activated by guanine nucleotide exchange factors (GEF), which.... : lines 67-68 in revision
line 84: Peripheral nerve injury induces changes in the expression....: line 92 in revision
line 290: ...axons in the dorsal column of the spinal cord.... line 326 in revision
line 308: The correlation between changes in gene expression and.......: line 347 in revision
line 447: Tissue homogenates were centrifuged at 12,000 × g for .....: line 522 in revision
line 448: …4oC, and the protein concentrations in the...: line 523 in revision
lines 449-450: Equal amounts of protein were resolved by SDS polyacrylamide gel electrophoresis, transferred to a nitrocellulose membrane, and ......: lines 524-525 in revision
line 456: Statistical analyses were performed using GraphPad Prism software.....: line 532 in revision
- To address the plagiarism issue, we have analyzed our manuscript by using two independent well-acknowledged plagiarism inspection programs, iThenticate and Turnitin. The inspection reports showed some similar expressions found from previous publications and, when we looked closely, we found that 1) the similarity was mostly detected in the phrases containing commonly used expressions for concepts of peripheral regeneration and βPix isoforms, or for widely used methodology such as western blots, and that 2) the similarities were mostly within 6 words in a row. Also, 3) the identified documents were mostly publications from us, supporting that the similarities were mainly caused by the use of personally preferred expressions by the authors. We regret that we were not careful enough to avoid repeated use of similar expressions. We thank the reviewer for raising the issue and have rewritten the sentences pointed out by the reviewer (highlighted in yellow in the revised manuscript).
6) Minor editing of English language required
- We used an English editing service to make sure the language is fine in the manuscript.
Reviewer 3 Report
Enabling a successful recovery following peripheral nerve injury relies on the critical reestablishment of essential connections required for effective communication within the nervous system through the process of axon regeneration. Within the present manuscript, the author demonstrates the significance of βPix as a crucial regulator. They highlights the overexpression of βPix in nerve segments containing regenerating axons, and its specific localization in the peripheral domain of the growth cone, suggesting a pivotal role in driving the extension of axons. By employing a genetic mouse model designed to selectively eliminate neuronal isoforms of βPix, the study underscores its contribution to the regrowth of neurites both in laboratory settings and in living organisms. Significantly, they find that the activation of cJun and STAT3 in cell bodies remains unaffected in mice with a βPix knockout underscores the localized impact of βPix within regenerating axons. The connection established between βPix and the activation of Src kinase further enhances the comprehension of the fundamental molecular pathways at play. Collectively, these findings offer valuable insights into the intricate coordination of axonal regrowth during the process of peripheral nerve regeneration.
Author Response
We are grateful for the opportunity to present a revised version of our manuscript “βPix Guanine Nucleotide Exchange Factor Regulates Regeneration of Injured Peripheral Axons”. The reviewers were enthusiastic about our manuscript and had constructive suggestions for improvements. We have addressed all of the comments of the reviewers, by revising Fig 1 and Fig 3, adding new supplementary figures (Fig S1 and S2), and extending our discussion to cover the issues raised by the reviewers. Fig 1 now has a new panel, Fig 1I, that shows a side-by-side comparison of the βPix isoforms in DRG neurons versus tissue. In Fig 3A, we included the injured βPix KO lane as requested. Fig S1 shows axonal regeneration one day after crush injury to monitor the time course of regeneration. In Fig S2, we show the p-Src levels in the sciatic nerve before and after injury. We believe that the revised manuscript is improved and hope that you and the reviewers will now find it to be appropriate for publication. A detailed response to individual reviewers is provided below (responses are in blue) and the manuscript was revised accordingly.
Reviewer 3
The reviewer was enthusiastic about our manuscript and thought that the findings in this study “offer valuable insights into the intricate coordination of axonal regrowth during the process of peripheral nerve regeneration”. We appreciate the reviewer’s comment. This reviewer did not raise any issue for revision.
Reviewer 4 Report
The study by Jeon et al. presents evidence supporting βPix's role in controlling growth cone formation and axon extension during regeneration. The findings demonstrate that the βPix protein, which experiences upregulation after nerve injury, localizes to both the regenerating axon shaft and the growth cone. Furthermore, the in vivo and in vitro experiments provide insights into the distribution of βPix neuronal isoforms during different stages of nerve regeneration, contributing to our understanding of the protein's role in axonal regrowth. The study concludes that increased βPix protein expression in the proximal stump of injured peripheral axons, particularly the neuronal isoforms βPix-b and βPix-d, contributes to axonal extension during peripheral nerve regeneration, likely through Src kinase-mediated regulation. Therefore, the authors suggest targeting βPix and its associated functions, such as Src kinase activity, could hold therapeutic potential for enhancing recovery from neural damage caused by trauma or neuropathic conditions. On this last point, I suggest the authors provide more evidence using samples from injured sciatic nerves to demonstrate that Src kinase activity correlates with Bpix regulation.
Major Comments:
· Figure 1 H. Provide loading control for this sample. In addition, this blot should show a side-by-side comparison of DRG neurons culture versus the DRG region from the mice sciatic nerve to validate that BPix-a is not an isoform in DRG neurons.
· Figure 3A. The three days after axotomy sample is missing for the BPix Niko model.
· To validate that Src kinase activity regulates Bpix localization in axonal growth, the authors can include a western blot analysis with samples used in experiment Figure 1F demonstrating total and phosphorylated Src levels. This is a critical experiment to perform, given that the authors suggest targeting Src kinase activity as a potential therapeutic avenue to promote axon regeneration in patients with nerve damage.
Minor Comments:
· Provide full name for BPix
· Page 2, Line 64: Provide full name for Rac1 and CDC42.
· Page 6, Line 179-182: Provide a reference citation for the previously generated BPix NIKO model.
· Figure 3 F. Indicate how many days after surgery the images represent in the figure legend.
· Provide the full name for Src Kinase and PP2 in the introduction or the result section 5.
Overall, the manuscript reads well, and the statements are clear.
Author Response
We are grateful for the opportunity to present a revised version of our manuscript “βPix Guanine Nucleotide Exchange Factor Regulates Regeneration of Injured Peripheral Axons”. The reviewers were enthusiastic about our manuscript and had constructive suggestions for improvements. We have addressed all of the comments of the reviewers, by revising Fig 1 and Fig 3, adding new supplementary figures (Fig S1 and S2), and extending our discussion to cover the issues raised by the reviewers. Fig 1 now has a new panel, Fig 1I, that shows a side-by-side comparison of the βPix isoforms in DRG neurons versus tissue. In Fig 3A, we included the injured βPix KO lane as requested. Fig S1 shows axonal regeneration one day after crush injury to monitor the time course of regeneration. In Fig S2, we show the p-Src levels in the sciatic nerve before and after injury. We believe that the revised manuscript is improved and hope that you and the reviewers will now find it to be appropriate for publication. A detailed response to reviewers is provided below (responses are in blue) and the manuscript was revised accordingly.
Reviewer 4
The reviewer mentioned that our study provides “evidence supporting βPix's role in controlling growth cone formation and axon extension during regeneration” and “insights into the distribution of βPix neuronal isoforms during different stages of nerve regeneration”. The reviewer did ask us to address several issues, especially to provide “more evidence to demonstrate that Src kinase activity correlates with βPix regulation”. We revised the manuscript following the suggestions and the revised text is highlighted in yellow in the revised manuscript.
Major Comments:
1) Figure 1 H. Provide loading control for this sample. In addition, this blot should show a side-by-side comparison of DRG neurons culture versus the DRG region from the mice sciatic nerve to validate that βPix-a is not an isoform in DRG neurons.
- We appreciate the suggestion and now provide the requested western blot results in revised Fig 1H (loading control) and new Fig 1I (side-by-side comparison of DRG neuron culture vs. DRG tissue). Please find the revised figures.
2) Figure 3A. The three days after axotomy sample is missing for the βPix NIKO model.
- We revised Fig 3A to include the fourth lane showing βPix NIKO three days after axotomy, as requested.
3) To validate that Src kinase activity regulates βPix localization in axonal growth, the authors can include a western blot analysis with samples used in experiment Figure 1F demonstrating total and phosphorylated Src levels. This is a critical experiment to perform, given that the authors suggest targeting Src kinase activity as a potential therapeutic avenue to promote axon regeneration in patients with nerve damage.
- To address this issue, first, we now cite a paper in which the authors showed that p-Src levels in sciatic nerves increased after crush injury, in both axons and Schwann cells (reference [48]). Second, to confirm that active Src is present during sciatic nerve regeneration, we performed experiments similar to what the reviewer suggested and included the results as a new supplementary figure Fig S2 (lines 277-278). We performed western blots to compare p-Src levels between control and crushed nerves at day 1 or 3 after injury. We divided the crushed nerves into three pieces, proximal, crush, and distal segments, to have a better spatial resolution. However, we did not observe a noticeable increase in p-Src levels after injury, but the p-Src levels appeared slightly downregulated three days after injury. Still, our results show that an active form of Src is present in injured nerve, consistent with its role in regulating βPix localization and axonal regeneration. Please see Fig S2 for the data. We regret that we could not perform western blotting for total Src because we could not obtain an antibody before the due of the revision.
Minor Comments:
1) Provide full name for βPix
- We included the full name, PAK-interacting exchange factor β, in the text (line 69) and in the new abbreviation/glossary list included as an appendix (page 15).
2) Page 2, Line 64: Provide full name for Rac1 and CDC42.
- We included the full name for CDC42, cell division cycle 42, in the text (line 65) and in the new abbreviation list (page 15). Rac1 is commonly known as Rac1 rather than its full name “Rac family small GTPase 1”, so we included the full name only in the abbreviation list (page 15).
3) Page 6, Line 179-182: Provide a reference citation for the previously generated βPix NIKO model.
- We thank the reviewer for this comment. We included a reference for the generation of the βPix NIKO model (page 7, reference [26]).
4) Figure 3 F. Indicate how many days after surgery the images represent in the figure legend.
- We wrote that the images were from three days after crush in the Fig 3F legend for clarity. We appreciate the comment.
5) Provide the full name for Src Kinase and PP2 in the introduction or the result section 5.
- We provided the full name for Src, proto-oncogene c-Src, in the text (line 85) and in the abbreviation/glossary list (page 15). PP2 is conventionally and commercially referred to as PP2. We described that PP2 is an inhibitor of the Src family of protein tyrosine kinases in the text (line 280) and also in the abbreviation/glossary list (page 15).
Overall, the manuscript reads well, and the statements are clear.
- We appreciate all the helpful comments.
Round 2
Reviewer 2 Report
I have no further suggestions.
Reviewer 4 Report
The authors have addressed the recommended suggestions.